# A 13-million turnover-number anionic Ir-catalyst for a selective industrial route to chiral nicotine

Congcong Yin[1,6], Ya-Fei Jiang[2,6], Fanping Huang[1], Cong-Qiao Xu [2], Yingmin Pan[1,3], Shuang Gao[1], Gen-Qiang Chen [1], Xiaobing Ding[4], Shao-Tao Bai [1,3] ✉, Qiwei Lang [4] ✉, Jun Li[2,5] ✉ & Xumu Zhang [1] ✉

Developing catalysts with both useful enantioselectivities and million turnover numbers (TONs) for asymmetric hydrogenation of ketones is attractive for industrial production of high-value bioactive chiral entities but remains a challenging. *Herein*, we report an ultra-efficient anionic Ir-catalyst integrated with the concept of multidentate ligation for asymmetric hydrogenation of ketones. Biocatalysis-like efficacy of up to 99% ee (enantiomeric excess), 13,425,000 TON (turnover number) and 224 s⁻¹ TOF (turnover frequency) were documented for benchmark acetophenone. Up to 1,000,000 TON and 99% ee were achieved for challenging pyridyl alkyl ketone where at most 10,000 TONs are previously reported. The anionic Ir-catalyst showed a novel preferred ONa/MH instead of NNa/MH bifunctional mechanism. A selective industrial route to enantiopure nicotine has been established using this anionic Ir-catalyst for the key asymmetric hydrogenation step at 500 kg batch scale, providing 40 tons scale of product.

In the context of industrial production of chiral entities estimated ca. >10,000 tons, development of practical catalysts with both excellent enantioselectivity and turnover numbers (TONs) for asymmetric hydrogenation is of great importance[1–4]. The celebrated Noyori-type catalysts often give useful enantioselectivities (>98% ee) but show few million turnover numbers (TONs) and a hundred turnover frequencies (TOFs) for asymmetric hydrogenation of aryl alkyl ketones[5]. In particularly, well-documented catalysts for challenging nitrogen-containing ketones that are relevant to construction of high-value bioactive compounds show at most 10,000 TONs, far from practical utilization (Supplementary Table 3 and Supplementary Fig. 3).

Compared to the representative hydrogenation catalysts[6–14] with inner-sphere mechanism, catalysts[15,16] of the outer-sphere mechanism can avoid the contact of substrates to the metal center, thus making it possible to design highly stable, enantioselective and coordination saturated hydrogenation catalysts (Fig. 1A). Indeed, Noyori-type catalysts, i.e., Ru(bisphos)(diamine) system by Noyori[17–19], Ir-PNN-complex by Zhou[20–23], and others[24–32] operating via NH/MH[19,27] bifunction mechanism are effective ketone asymmetric reducing catalysts (Fig. 1A, Supplementary Fig. 2, Supplementary Information 3.1 and Supplementary Table 2). Exceptionally, Zhou's tridentate ligand of Ir-PNN-catalyst provides the record-high TONs of 4,550,000 at 98% ee (enantiomeric excess).

Given the increasing capital investments, operational costs and sustainability requirements, to obtain ultra-efficient catalysts achieving 10-million TONs and biocatalysis-like reaction turnover frequencies (TOFs), beyond the state-of-the-art NH/MH bifunction catalysts, is highly important and has been the holy grail of the

[1]Department of Chemistry, Academy for Advanced Interdisciplinary Studies and Shenzhen Grubbs Institute, Southern University of Science and Technology, Shenzhen 518055, China. [2]Department of Chemistry and Guangdong Provincial Key Laboratory of Catalytic Chemistry, Southern University of Science and Technology, Shenzhen 518055, China. [3]Center for Carbon-Neutrality Catalysis Engineering and Institute of Carbon Neutral Technology, Shenzhen Polytechnic, Shenzhen 518055, P. R. China. [4]Shenzhen Catalys Technology Co., Ltd, Shenzhen 518100, China. [5]Department of Chemistry and Engineering Research Center of Advanced Rare-Earth Materials of Ministry of Education, Tsinghua University, Beijing 100084, China. [6]These authors contributed equally: Congcong Yin, Ya-Fei Jiang. ✉e-mail: shaotaobai@szpt.edu.cn; qwlang@catalys.com.cn; junli@tsinghua.edu.cn; zhangxm@sustech.edu.cn

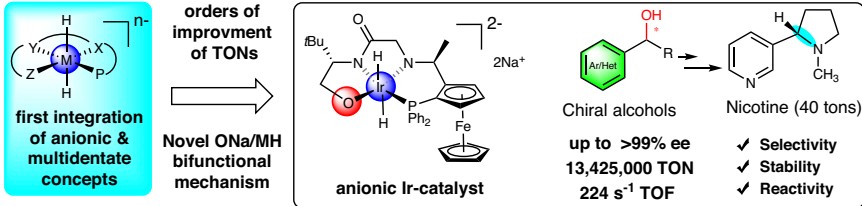

**A** State-of-the-art Noyori-type catalysts (outer-sphere mechanism) for asymmetric hydrogenation of ketones

Noyori (1998)
80% ee, 2,400,000 TON, TOF 63 s⁻¹

Zhou (2011)
98% ee, 4,550,000 TON

Zhang (2017)
>99% ee, 1,000,000 TON

**B** Conceptually advanced ultra-efficient anionic Ir-catalyst for asymmetric hydrogenation of ketones

first integration of anionic & multidentate concepts

orders of improvment of TONs

Novel ONa/MH bifunctional mechanism

anionic Ir-catalyst

Chiral alcohols
up to >99% ee
13,425,000 TON
224 s⁻¹ TOF

Nicotine (40 tons)

✓ Selectivity
✓ Stability
✓ Reactivity

**Fig. 1 | Development of practical catalysts for asymmetric hydrogenation of ketones affording high-value chemicals. A** The state-of-the-art neutral Noyori-type catalysts for asymmetric hydrogenation of ketones. **B** Our conceptually advanced ultra-efficient anionic Ir-catalyst for asymmetric hydrogenation of (nitrogen-containing) ketones. ee: enantiomeric excess.

homogeneous transition metal catalysis[33–35]. Inspired by highly reactive anionic reductants and multidentate Noyori-type hydrogenation catalysts, we proposed the integration of the concepts of anionic complexes and multidentate ligands for developing ultra-efficient asymmetric hydrogenation catalysts with high selectivity, stability, and reactivity preeminence (Fig. 1B). The characteristic anionic complexes bearing a formal negative charge can, in principle, enable high hydricity[36] and accordingly high catalytic reaction rates, as evidenced from the seminal anionic metal hydride catalysts for hydrogenation reaction of carbonyl compounds by Pez[37–39], Poli[40], and others[40]. Importantly, compared to traditional anionic catalysts, multidentate ligands should help to stabilize the metal center through coordinative saturated 18 electron complexes and form well-defined chiral environment for converting specific substrates and give rise to highly stable and selective catalysts.

In this manuscript, we report a simple and easily fabricated tetradentate ligand f-phamidol-based anionic Ir-catalyst that provides unprecedented up to 13 million TONs, hundreds of TOFs per second at ultra-high selectivity (>99% ee, 13,425,000 TON and 224 s⁻¹ TOF, Fig. 1B, and Supplementary Tables 2, 3) comparable to biocatalysts. With this catalyst, even ketones with awkward coordinating basic nitrogen have been firstly hydrogenated at a million TON and >99% ee selectivity, resulting in a novel selective industrial route to enantiomeric pure nicotine, already providing 40 tons of product. Importantly, significantly improved hydricity and a novel ONa/MH bifunctional mechanism are presented based on a combination of in situ spectroscopy studies, DFT calculations and reaction kinetic experiments.

## Results
### Catalytic reaction optimization
Initially, on the basis of our previous work[22,26,41], iridium catalysts based on tetradentate PNNO ligand f-phamidol was examined at a substrate/catalyst ratio of 2,000,000 using 80 mmol benchmark acetophenone at ambient temperature (Supplementary Tables 4–6). All the base gave exceptionally high enantioselectivities >99% ee, whereas the highly basic NaOtBu that leads to 99% conversions in 16 h (corresponding to 1,980,000 TONs) is superior to the others with variation of either the alkali ions or anionic counterions. Compared to neat experiments, the addition of solvent is important for achieving both high

enantioselectivity and TON. Under the optimal conditions, high turnover experiments with a variation of the substrate/catalyst ratio from 5,000,000 to 15,380,000 were carried out. Remarkably, a steady increasing TONs from 4,835,000 to 11,535,000 was observed at excellent enantioselectivities of 99% ee and excellent conversions of 75–97%, indicating super-stable, durable, and enantioselective Ir/f-phamidol catalyst at the highest loading of substrate (*vide infra*, Fig. 1A, Supplementary Table 6). Upon increasing the substrate amount to 800 mmol at 100 bar H₂, the highest TON of 13,425,000 together with 89.5% conversion and 99% ee were observed in a 30-days' reaction, implying outstanding averaged production rate of 2846 kg_product (kg_precatalyst)⁻¹ h⁻¹. The initial TOFs were calculated based on a pressure-drop curve (Supplementary Fig. 34 and Supplementary Information 5.1). An ultra-high initial TOF of 224 s⁻¹ was recorded, which is close to the biocatalytic efficiency of (de)hydrogenase[42–44].

### Evaluation of substrates and scale-up utilization
Given the successful application of the Ir/f-phamidol catalyst for asymmetric hydrogenation of benchmark acetophenone, we further examined its efficiency in a more challenging conversion of nitrogen-containing ketones as well as in laboratory scale-up construction of enantiomeric pure nicotine (Fig. 2, Supplementary Fig. 3, Supplementary Table 3, Supplementary Information 2.5-6). Ketone **S2** containing an amide function proceeds smoothly to afford the desired chiral compound at gram-scale in >99% ee, 97% conversion and 970,000 TONs, with orders of improvement compared to previous work (Supplementary Fig. 3 and Supplementary Table 3). Remarkably, even though ketone **S3** contains both an amide function and a pyridine function that often results in catalysts deactivation, our catalyst still gave exceptionally high efficiency (>99% conv., >99% ee, and 1,000,000 TONs) at gram scale. Remarkably, substrate **S4** can be also successfully converted at 40 kg scale at substrate catalyst ratio of 60,000. Utilization the asymmetric hydrogenation reaction as a key step, a novel laboratory scale route to chiral Nicotine was established, affording chiral Nicotine in 99% ee in 68% yields in three steps from readily available substrate **S4**.

Nicotine, generally isolated from tobacco, is one of the most important bioactive natural products with estimated consumption of > 1000 tons per year with the tendency of steady increasing. Therefore, based on laboratory data, the Ir/f-phamidol catalyst was

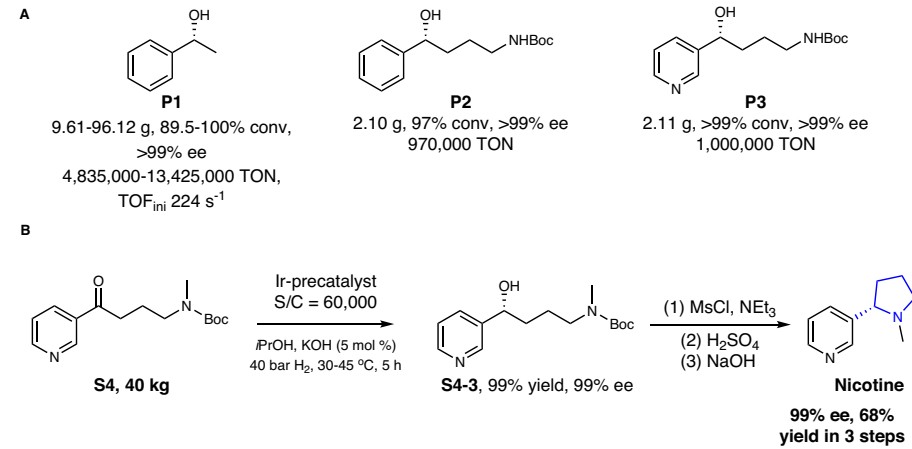

**Fig. 2 | Asymmetric hydrogenation of ketones using Ir/phamidol catalyst.** **A** Laboratory scale asymmetric hydrogenation of representative benchmark acetophenone and nitrogen-containing aromatic ketones. **B** Laboratory scale-up asymmetric hydrogenation of nitrogen-containing ketone as a key step in a novel selective route to chiral nicotine.

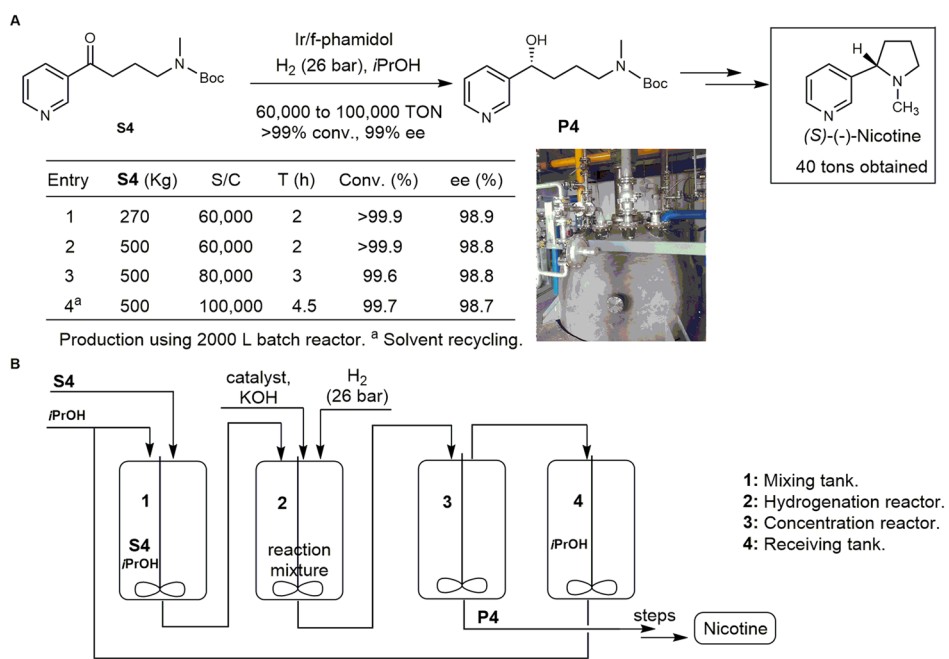

**Fig. 3 | A selective industrial route to chiral nicotine.** **A** Batch process data of the key asymmetric hydrogenation of **S4**; **B** Schematic of our developed process for manufacture of chiral nicotine.

investigated in a selective industrial route to enantiomeric pure nicotine from easily available ketone **S4**. A 2000 L continuously stirred tank reactor was used for the crucial asymmetric hydrogenation, shown in Fig. 3A, B. Given economy and safety concerns, 26 bar dihydrogen pressure and much cheaper KOH instead of NaOtBu were applied based on conditions optimization (Supplementary Table 4). Stepwise scale up the feedstock from 270 to 500 kg at substrate/catalyst ratio of 60,000 gave full conversions, 98.9% ee in 2 h (Fig. 3A, entries 1–2). Increasing the substrate/catalyst ratio to 80,000 and 100,000 at 500 kg scale needs 3 h and 4.5 h, respectively, and the reaction reached 99.6–99.7% conversion with 98.7–98.8% ee (Fig. 3A, entries 3 and 4). To our surprise, a substrate/catalyst ratio of 100,000 at 500 kg scale using recycled solvents can be operated smoothly, giving rise to a key chiral alcohol intermediate (99.7% conversions, 98.7% ee, 100,000 TON, with catalyst load 0.045 g kg⁻¹ product and space-time-yield 55.6 g L⁻¹ h⁻¹) for production of nicotine already 40 tons with 99% ee.

## Characterization of catalyst

To understand the unique properties of the robust catalyst, we performed detailed characterization of the Ir/f-phamidol catalyst via a combination of experimental techniques and quantum-theoretical modeling based on density functional theory (DFT, Supplementary Information 2.7). Upon mixing f-phamidol with [Ir(COD)Cl]₂, monochloride dihydride iridium complexes were identified as the Ir-precatalyst (Fig. 4, in dashed box, Supplementary Figs. 4–16), based on the evidences from HRMS, NMR, ATR-IR, Raman, XRD and DFT calculations. Briefly, DFT calculations, NMR and ATR-IR spectroscopy confirmed that the formation of mixtures of Ir-precatalysts with *cis*-configuration of hydride and carbonyl binding to Ir-metal (CO-bind *cis*) is slightly favorable. Two *cis*-hydrides instead of *trans*-hydrides are evidenced by ATR-IR analysis of the dry powder at 2229 and 2127 cm⁻¹ and DFT modeling (Supplementary Figs. 7, 8). The evidence of amide-carbonyl instead of NH coordination to Ir-metal was supported by the red shifts up to 22 cm⁻¹ of the amide-carbonyl group. Upon

**Fig. 4 | Characterization of Ir-complexes.** Formation of anionic Ir-catalyst (NH-bind *cis* and CO-bind *cis*, in solid box) from Ir-precatalyst (**C** and **D**, in dash box) through complex **A** as evidenced by HRMS, NMR, IR, Raman, XRD, DFT, and catalysis data. Calculated relative Gibbs free energies at 298.15 K are given in brackets.

**Fig. 5 | Performance of modular modified ligands in comparison with f-phamidol.** Enantioselectivities and turnover numbers are given under the ligands.

deprotonation of the alcohol donor of the Ir-precatalyst, favorable tetra-coordinated dihydride Ir-ate complexes (Fig. 4, in solid box, Supplementary Figs. 17–24) were formed via likely intermediate neutral complex **A**. Compared to deprotonation of NH function, OH deprotonation is highly energetically favorable, resulting in anionic Ir-catalysts **C** and **D** that are likely under equilibrium at the experimental condition. Such anionic Ir-complexes were characterized by HRMS, in situ high-pressure (HP) NMR and ATR-IR spectroscopy and DFT calculations. HRMS of a solution of Ir-precatalyst and NaO*t*Bu in isopropyl alcohol under 30 bar $H_2$ showed exact mass of 765.1874 $[M-2Na + 3H]^+$ and 799.1490 $[M-2Na + 2H+Cl]^-$, corresponding to protonated and chlorinated anionic Ir-catalysts, respectively, in the positive and negative region. The existence of amide-N instead of amide-carbonyl coordination to Ir-metal was supported by blue shifts observed by in situ high pressure (HP) ATR-IR experiments and DFT calculated infrared spectra (Supplementary Figs. 22, 23).

To confirm the crucial role of tetradentate PNNO ligand for the formation of ultra-efficient anionic Ir-catalyst, we prepared slightly modified ligands for asymmetric hydrogenation of acetophenone as references (Fig. 5, Supplementary Fig. 25). Ligand f-phamidol-N-Me with NH function being methylated gave a maximum 100,000 TON and 95% ee. Ligand f-phamidol-O-Me with OH function being methylated displayed significantly dropped TON of 33,000 and 76% ee, implying the decisive role of the extra anionic oxygen donor in creating the robust TON, reaction rate, and selectivity. Despite the presence of both NH and OH, ligand f-phamidol-Nacl-Me, which cannot form tetra-coordinated anionic Ir-catalysts due to unlikely amide-NH coordination to Ir-metal originated from both steric and electronic reasons, showed a rather small TON of 7000 and 35% ee. Those control experiments provide unequivocal evidence that the OH functional of tetradentate PNNO ligand plays a critical role in asymmetric hydrogenation of acetophenone, which is different from traditional NH/MH bifunctional catalysts.

## Mechanistic study

Quantum chemistry studies and reaction kinetics experiments were performed to address the mechanism of the ultra-efficient anionic Ir-catalyst (see Supplementary Information 5, Supplementary Data 1 and Supplementary Figs. 32–47). Upon deprotonation, Ir-complex **B** is significantly higher in energy (ca. >20 kcal mol$^{-1}$) than the active anionic Ir-complexes (*vide supra*, Fig. 4, Supplementary Fig. 24). We therefore used the anionic Ir-complexes **C** and **D** in comparison with **A** as starting structures for DFT calculations. The well-known NNa/MH bifunctional mechanism[45] and a novel ONa/MH bifunctional mechanism were all carefully explored. Compared to **A** and anionic Ir-catalyst **C** (Supplementary Figs. 26, 27, Supplementary Table 7), anionic Ir-catalyst **D** provides energetically favorable pathways for both NNa/MH and ONa/MH bifunctional mechanisms, as shown in Fig. 6. The alkali cation (Na$^+$) can polarize the carbonyl of the ketone substrate and thus facilitate hydride transfer in the selectivity determining step from the anionic Ir-catalyst to form the alkoxide intermediate **III**, which is smoothly converted to alcohol product and the initial active catalyst upon taking a dihydrogen molecule. Importantly, the ONa/MH bifunctional pathway giving the desired enantiomer has a much lower free energy barrier than that of the NNa/MH bifunctional pathway (*viz.* 6.2 vs 10.1 kcal mol$^{-1}$), consistent with our experimental observations. Additionally, when explicit solvent molecules were considered in the models to simulate the actual solvated cations and hydrogen bonding environment (Supplementary Fig. 28), the predicted free energy barrier of 12.9 kcal mol$^{-1}$ is comparable to the experimental TOF according to the Arrhenius equation. reaction Interestingly, preliminary kinetics experiments reveal, unexpectedly, 1.9 order in dihydrogen pressure

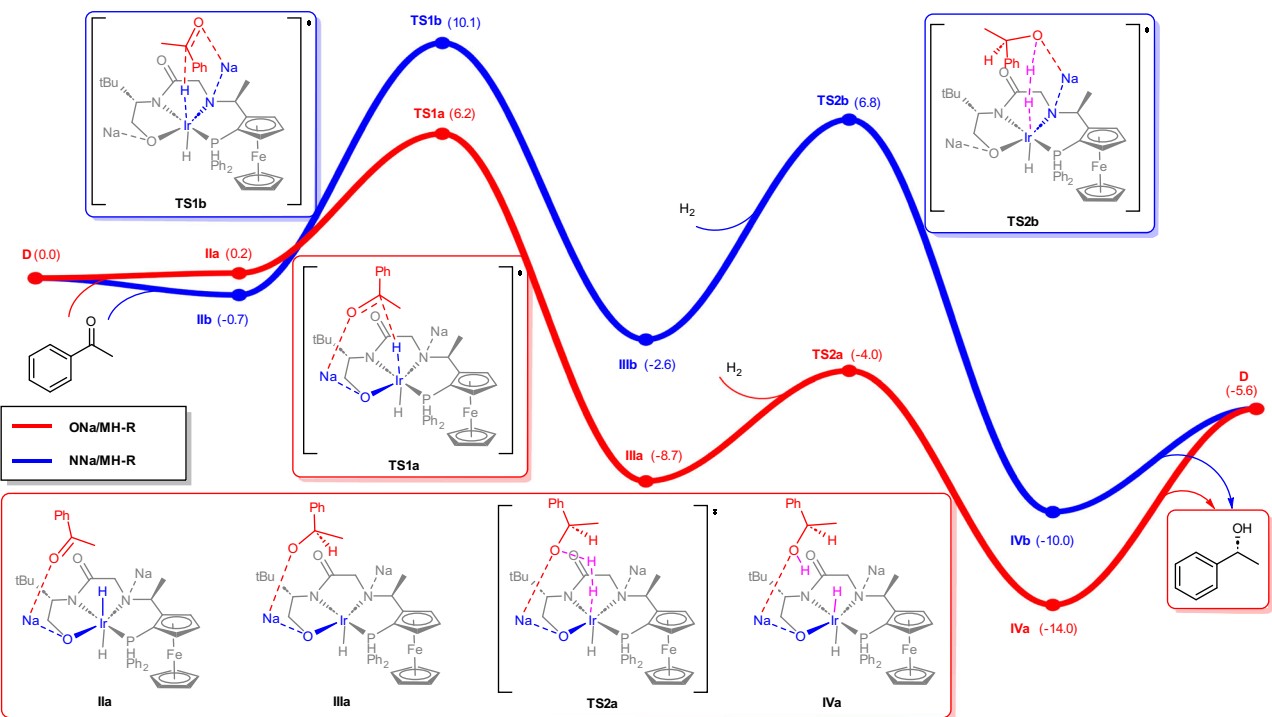

**Fig. 6 | DFT calculations on the mechanism.** Predicted Gibbs free energy profile for the asymmetric hydrogenation of acetophenone via the active anionic Ir-catalyst **D** through ONa/MH bifunctional (in red) and NNa/MH bifunctional (in blue) paths.

and 1.5 order in iridium concentration (Supplementary Information 5, Supplementary Table 10 and Supplementary Figs. 32–47), implying the rather complexity of this catalyst deserving further research.

The origin of the significantly enhanced activity upon introduction of anionic donor can be rationalized by the orbital interactions between the ligation-tunable $5d$ orbitals of Ir atom and $1s$ orbital of hydride (Fig. 7). The f-phamidol ligand in tetradentate coordination manner stabilizes the Ir-ate catalyst (18 electron complex) under basic condition and the anionic donor greatly elevates the $5d$-orbital energy of Ir atom. Higher $5d$-orbital energy level of Ir atom leads to weaker orbital mixing with $1s$ orbitals of axial hydride based on Pimentel–Rundle three-center-four-electron (3c-4e) model[46,47], which causes larger composition of H $1s$ orbitals in the bonding/nonbonding orbitals and subsequently larger electron density on hydride atoms as well as stronger hydricity of the catalysts. The Ir atom and two axial hydrides in the Ir-ate catalyst forming 3c-4e bonding is confirmed by DFT and the ab initio complete active space self-consistent field (CASSCF) calculations (Supplementary Figs. 29–31), where the natural orbital occupation numbers (NOONs) of the three orbitals formed by $5d_{z^2}$ orbital of Ir atom and $1s$ orbitals of hydrides are 1.99, 1.97, 0.03, respectively.

## Discussion

In summary, we presented an anionic Ir-catalyst for asymmetric hydrogenation of acetophenone with record-high turnover-numbers of 13-million, $224\,s^{-1}$ TOF and >99% ee and challenging nitrogen-containing ketones with unprecedented 1 million TONs and 99% ee as a key step for a novel industrial route to chiral nicotine. HRMS, in situ HP ATR-IR, HP NMR, Raman, XRD characterization and quantum-theoretical modeling and as well reaction kinetics studies demonstrate highly improved metal-hydride hydricity and a novel ONa/MH bifunctional mechanism. This work will likely inspire the development of other ultra-efficient homogeneous anionic catalysts for production of chemicals, fuels and materials.

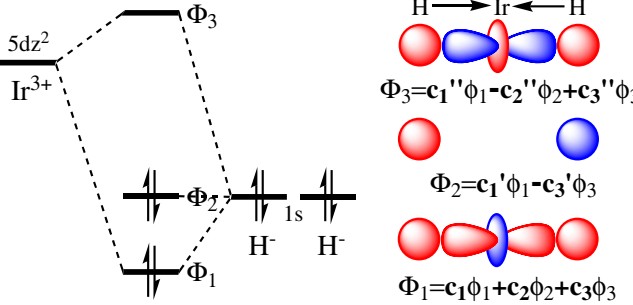

**Fig. 7 | Theoretical origin of the significantly enhanced activity.** Schematic three-center-four-electron (3c-4e) orbital interactions between Ir and the hydride atoms.

## Methods

### Optimal conditions for 10 million turnover number experiments (at S/C = 15,000,000)

To a 20.0 mL vial was added the precatalyst (3.2 mg, $4.0 \times 10^{-3}$ mmol) and anhydrous $i$PrOH (10.0 mL) in an argon-filled glovebox. The mixture was stirred for 0.5 h at 25 °C. And then 800 mmol of acetophenone and NaO$t$Bu (96 mg, 1 mmol) were added into a 300 mL hydrogenation vessel. Then 20 mL anhydrous $i$PrOH was added and a solution of Ir-precatalyst in anhydrous $i$PrOH (133 μL) was added via an injection port. Then the vessel was placed in an autoclave, which was closed and moved out from golvebox. The autoclave was quickly purged with hydrogen gas for three times, and then pressurized to 100 bar H$_2$ (keeping the hydrogen pressure not lower than 80 bar). The reaction solution was stirred at room temperature until for 30 d, and then the pressure was released carefully. The solution was removed under reduced pressure. Conversion was determined by $^1$H NMR analysis, and ee was determined by HPLC with a chiral stationary phase. 89.5% conv., 99% ee, TON = 13,425,000.

## Applied asymmetric hydrogenation procedure for construction of Nicotine at 40 kg scale

Under nitrogen atmosphere, to a 20.0 mL vial was added Ir-precatalyst (1.92 g, 2.4 mmol) and anhydrous *i*PrOH (10.0 mL). The mixture was stirred for 0.5 h at 25 °C. In a 200 L Hastelloy hydrogenator was charged with 40 kg compound **S4** (144 mol) in 80 L isopropanol at room temperature. 400 g KOH (7.14 mol, 5 mol %) was added to the reactor and the resulting solution was degassed by five cycles of vacuo followed by filling with nitrogen. The previously prepared solution of catalyst (S/C = 60,000) in *i*PrOH was transferred to the hydrogenator under a stream of nitrogen by cannula. Hydrogen was initially introduced into the autoclave at a pressure of 40 bar. The reaction mixture was stirred while maintaining a temperature range of 30–45 °C, monitored by hydrogen consumption and HPLC. The reaction was complete after 5 h. The reaction mixture was cooled to 25 °C, and hydrogen was replaced by nitrogen. The solution was transferred to a glass-lined reactor and concentrated in vacuo. Crude compound **S4-3** was obtained as red oil: 41 kg (99% yield, 99% ee). The crude compound **S4-3** was used in the next step without further purification.

A 10-L, 4-neck round-bottom flask was equipped with a mechanical stirrer, a condenser with an N$_2$ inlet, a thermowell, and an addition funnel. The flask was charged with 400 g of compound **S4-3** (1.43 mol, crude product), 232 g triethylamine (2.3 mol, 1.6 equiv.), and 2.3 L of MTBE at room temperature. The resulting reaction mixture was cooled to −10 °C, followed by dropwise addition of 229 g methanesulfonyl chloride (2 mol, 1.4 equiv.). The reaction was allowed to stir at −5 °C for 2 h. A pale-yellow suspension was observed. After that, the insoluble solids were filtered off and the solids was rinsed with MTBE (2 × 500 mL). The combined organic phase was washed by saturated aqueous NaHCO$_3$ (2 × 1 L). The reaction mixture was cooled to −10 °C, followed by dropwise addition of 1.86 kg sulfuric acid aqueous solution (30% by weight, 4.0 equiv.). The mixture was then allowed to gradually warm to room temperature and stirred for 2 h. The organic phase was separated to waste and the lower aqueous phase was recharged to the reactor. The reaction mixture was allowed to cool to −10 °C. Finally, 7 L of sodium hydroxide solution (2 M) was added slowly until the pH reached 10 - 11. The aqueous phase was extracted with ethyl acetate (3 × 5 L) at room temperature. The combined organic solvent was evaporated under reduced pressure and the resulting residue was further purified by distillation under vacuum at 70 °C to afford pure product **(S)-Nicotine** as colorless oil (157 g, 0.97 mol, 68% yield in 3 steps, 99% ee). [α]$_D^{25}$ = −107.3 (c = 1.00 in CHCl$_3$). The enantiomeric excess was determined by HPLC on Chiralcel OD−3 column, 254 nm, 30 °C, *n*-hexane (0.1% DEA): *i*PrOH = 95:5; flow rate 1.0 mL/min; t$_R$ (major) = 5.6 min, t$_R$ (minor) = 6.3 min. $^1$H NMR (400 MHz, CDCl$_3$) δ 8.54 (d, $J$ = 1.8 Hz, 1H), 8.49 (dd, $J$ = 4.8, 1.6 Hz, 1H), 7.71 (dt, $J$ = 7.8, 1.8 Hz, 1H), 7.26 (dd, $J$ = 7.8, 4.8 Hz, 1H), 3.27 – 3.22 (m, 1H), 3.09 (t, $J$ = 8.3 Hz, 1H), 2.31 (q, $J$ = 9.2 Hz, 1H), 2.23 – 2.18 (m, 1H), 2.16 (s, 3H), 2.02 – 1.91 (m, 1H), 1.87 – 1.68 (m, 2H). $^{13}$C NMR (101 MHz, CDCl$_3$) δ 149.2, 148.3, 138.4, 134.6, 123.3, 68.6, 56.7, 40.1, 34.9, 22.3. HRMS (ESI) Calculated for C$_{10}$H$_{15}$N$_2$ [M + H]$^+$ 163.1235; found 163.1231.

## Data availability

The data supporting the findings of this study are available within the paper and its Supplementary Information. All other data are available from the corresponding author upon request.

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

## Acknowledgements
This work was supported by National Key R&D Program of China (grant 2021YFA1500200), National Natural Science Foundation of China (grant 22033005, 22038002, and 21991113), Stable Support Plan Program of Shenzhen Natural Science Fund (grant No. 20200925161222002), Shenzhen High-Caliber Personnel of SZPT (6023330003K) and Guangdong Provincial Key Laboratory of Catalysis (grant No. 2020B121201002). Computational resources are supported by the Center for Computational Science and Engineering (SUSTech) and Tsinghua National Laboratory for Information Science and Technology.

## Author contributions
X.Z., J.L., S.T.B., and Q.L. conceived the idea and directed the project. C.Y. designed and conducted the experiments with inputs and support from Q.L., F.H., S.G., X.D., and G.Q.C. Y.F.J. performed all the theoretical computations with guidance of J.L. and C.Q.X. S.T.B. performed the characterizations with support from Y.P. using HRMS, in situ HP ATR-IR, HP NMR, Raman, XRD spectroscopy. S.T.B. and Y.F.J. analyzed the data and drafted the manuscript. All authors contributed to the manuscript.

## Competing interests
X.D., S.G., and Q.L. are inventors on patents (WO 2021/212880 Al, EP 3 925 955 A1, CN 113527187A, US 2022/0089564 Al, CN 202210771902.X, and 202111097411.3), held and submitted by Shenzhen Catalys Technology Co., Ltd. Other authors do not have competing interests.
