## [Peer Review File · Nature Communications]

REVIEWER COMMENTS

Reviewer #1 (Remarks to the Author):

The paper by Yin et al. describes the use of a double anionic iridium ate complex for the asymmetric hydrogenation of aromatic ketones and aromatic ketones containing nitrogen functionalities. The product alcohols were obtained with 99% ee, but the most spectacular aspect of this work are the extremely high turnover numbers and turnover frequencies. A mechanism is proposed that was supported by control experiments and with DFT calculations. The anionic nature of the catalyst leads to an increased hydricity of the iridium hydride and at the same time the carbonyl group is activated by coordination to the sodium on the alkoxy group of the ligand. These results are very interesting and will appeal to a broad readership.

The synthetic aspects of the paper as well as the large-scale application are fine. However, I see some issues in the mechanistic part that will need further work.

Looking at the DFT calculations it becomes clear that the transfer of hydride to the ketone is the rate-determining step in the red mechanism which is based on involvement of the sodium alkoxide. On the contrary, in the blue mechanism the reaction of the complex with hydrogen is the rate determining step. Thus, it would stand to reason to determine the order of the reaction in hydrogen in order to gain additional experimental evidence for this mechanism.

Determining the reaction order in iridium is also necessary, as this will show if the iridium complex that is actually involved in the hydrogenation is dimeric or not. Thus, I suggest the authors do a full kinetic study to determine the order in hydrogen, iridium and substrate.

The authors do show a hydrogen consumption curve in the supp info. However there is a blue as well as a red curve and the authors do not explain what the difference is between the two.

The effect of the amount of solvent is rather bizarre as the product is also a secondary alcohol. Are the authors 100% sure that this effect is not caused by a minor impurity in the isopropanol? Isopropanol can even contain peroxides that could well have caused this effect. It may be worthwhile to repeat these experiments with rigorously purified isopropanol.

The stability of this catalyst is indeed remarkable. Is it possible to reuse the spent catalyst after a first reaction?

What is entirely lacking in the manuscript is a summary of the state-of-the-art of the use of anionic complexes in (asymmetric) hydrogenation. Pez already reported the use of anionic complexes in homogeneous hydrogenation of esters in 1981. Then there is of course the work of Fujita, of Rinaldo Poli, and I am sure there will be others as well.

The interesting thing about the Pez work is that other later showed that the anionic complexes were readily protonated by the alcoholic solvent. Are the authors 100% sure that this is not also the case here. It would be interesting to measure a MS of the catalyst after or better during hydrogenation to see if the catalyst is still double anionic.

Some minor points.

Use of acronyms in the title is usually not allowed.

I also do not see the added value of the use of this triple S. I would recommend to remove this entirely from the manuscript.

I would also remove the free commercial for Nicotinell. Most journals do not allow this.

After these major revisions, the paper can likely be published.

Reviewer #2 (Remarks to the Author):

In this manuscript, Zhang and coworkers reported a highly efficient asymmetric hydrogenation of ketones catalyzed by an unprecedented Ir-ate catalyst. One of the most impressive aspects is the more than 10 million TON. This is the highest number reported in the field to date, comparable to biocatalytic hydrogen transfer. The reasons for the high activity were analyzed and the ate type Ir complex was found to be the key factor to achieve the excellent performance. This is also an important discovery. On the other hand, this strategy and method have also been used for the efficient synthesis of important chiral compounds including nicotine, and 40 tons of nicotine have

been produced, which is also a remarkable achievement. The paper is well prepared, including the relevant theoretical calculation part. Therefore, in view of the innovation and application value of this work, I recommend this work to be published in Nature Communications.

In addition, some other literatures on highly efficient asymmetric hydrogenation of ketones catalyzed by different transition metals are recommended to be cited, such as: (Ni) *Angew. Chem. Int. Ed.* 2022, 61, e202115983; (Ir) *Nat. Catal.* 2020, 3, 621–627; (Mn) *Angew. Chem. Int. Ed.* 2019, 58, 4973–4977; (Ir) *Org. Lett.*, 2018, 20, 6135–6139; (Pd) *Angew. Chem. Int. Ed.*, 2013, 52, 11632–11636.

Reviewer #3 (Remarks to the Author):

This communication reports on the performance of an iridium catalyst for asymmetric hydrogenation. According to this work, this is already being used in an industrial process for the synthesis of nicotine. The work is closely related to prior work by the authors (cited as refs. 23, 41 and 42, where the last one looks like a pre-print of the present communication) and it is not entirely clear to me what the focus of this manuscript is meant to be - the catalyst development, the industrial process or the accompanying mechanistic postulate and its analysis. I'm thus wondering whether this is of sufficient novelty and significance to the field, and also whether a communication is the correct format - the ESI is extensive and, in my view, necessary to make the paper accessible, such that I would favour presentation as a full paper.

The writing is not very clear, with distracting errors of grammar and expression. The abstract is not written in the right style, contains multiple references and some of the information is then repeated in the introduction. In addition, abbreviations are used extensively, affecting the clarity of the writing, and figures appear overloaded and are difficult to follow. These presentational issues distract from the science and will need to be addressed.

The reported work is based on multiple experiments and calculations, but these have not been presented well or logically, making it look like a set of results has been cherry-picked for the communication, while the details have been buried in the ESI. For example, I wondered about the small energy differences calculated between the mono- and diprotonated versions of the catalyst and how these figures were balanced with the sodium base; this has fed into figure 4B, but the details/critical assessment of the results are hard to come by.

Similarly, some tests with different numbers of explicit solvent molecules seem to have been performed and are shown in Figure S38, but I struggled to locate a discussion of the different models, how solvent molecules were placed and, again, a critical assessment of the results.

I also wondered whether for this type of study, optimisation in the gas phase followed by calculations of energies in solvation, with a bigger basis set, would be appropriate - at the very least the authors would need to demonstrate that the structural effects are minimal (and I fear they would not be).

Calculation results have mainly been presented in terms of energy profiles in the ESI, again with minimal commentary on what was done and why - it would be sensible to expand this into data tables, discussions and the critical evaluations of whether these results can be trusted. For ee's the energy differences are small, such that computational and conformational noise need to be assessed carefully and I find it difficult to fully assess whether the interpretations of these results are reliable.

Overall, then, there is a lot of work here, but the presentation needs to be improved and the key messages need to be clarified - I'm not convinced this is the right format or indeed the right journal.

REVIEWER REPORT
COMMENTS TO AUTHOR:

Reviewer 1: *The paper by Yin et al. describes the use of a double anionic iridium ate complex for the asymmetric hydrogenation of aromatic ketones and aromatic ketones containing nitrogen functionalities. The product alcohols were obtained with 99% ee, but the most spectacular aspect of this work are the extremely high turnover numbers and turnover frequencies. A mechanism is proposed that was supported by control experiments and with DFT calculations. The anionic nature of the catalyst leads to an increased hydricity of the iridium hydride and at the same time the carbonyl group is activated by coordination to the sodium on the alkoxy group of the ligand. These results are very interesting and will appeal to a broad readership.*

The synthetic aspects of the paper as well as the large-scale application are fine. However, I see some issues in the mechanistic part that will need further work.

Response: Thanks for your remarks and kind recommendation and as well constructive suggestions. We have thoroughly revised the manuscript and supplementary information with additional reaction kinetics experiments in dihydrogen pressure and iridium concentration. All the corrections were highlighted in yellow color.

Corrections suggested are as follows.

Looking at the DFT calculations it becomes clear that the transfer of hydride to the ketone is the rate-determining step in the red mechanism which is based on involvement of the sodium alkoxide. On the contrary, in the blue mechanism the reaction of the complex with hydrogen is the rate determining step. Thus, it would stand to reason to determine the order of the reaction in hydrogen in order to gain additional experimental evidence for this mechanism.

Response: Thanks for this important notification. We apologize for a numerical mistake: After careful examination of the original data, we find a mistake in use of D' instead of D'' as reference in calculating the relative Gibbs free energy of IIIb. The correct value should be -2.6 kcal mol⁻¹ instead of -6.7 kcal mol⁻¹ in the blue mechanism. Figure 5 (shown below) and Figure S26 have also been corrected. All the original energy data are summarized in Table S11-S14 and the cartesian coordinates of all optimized structures are listed in Section 10 in the supporting information. Therefore, the transfer of hydride to acetophenone is indeed the rate/enantioselectivity-determining step for both the red mechanism and the blue mechanism. We truly appreciate your careful review that have helped us to avoid this mistake! Thanks so much!!

Fig. 5 Predicted Gibbs free energy profile for the asymmetric hydrogenation of acetophenone via the active anionic Ir-catalyst **D** through ONa/MH bifunctional (in red) and NNa/MH bifunctional (in blue) paths.

Importantly, for the energetically preferred red mechanism, the Gibbs free energy barriers of the hydride transfer step and dihydrogen activation step are 6.2 and 4.7 kcal mol⁻¹, respectively, that are of small differences. In line with this, reaction kinetics experiments show 0.5 order in dihydrogen pressure, implying that the activation of dihydrogen is partaken in the determination of the overall reaction kinetics. The data of the kinetics experiments were presented in Table S10 and Figure S32-39 with experimental details in SI 5.2. A conclusion of the reactions kinetics experiments is added in the main text as follow:

“The comparable Gibbs free energy barriers of the hydride transfer step and dihydrogen activation step are also consistent with kinetic experiments (SI 5, Table S10 and Figs. S32-47) where 0.5 order in dihydrogen pressure and 1.5 order in iridium concentration were observed.”

Determining the reaction order in iridium is also necessary, as this will show if the iridium complex that is actually involved in the hydrogenation is dimeric or not. Thus, I suggest the authors do a full kinetic study to determine the order in hydrogen, iridium and substrate.

Response: Reaction order in iridium is done. A 1.5 order in iridium was observed, implying monomeric iridium ate catalyst is operation under catalysis conditions. The data of the kinetics experiments were presented in Figure S40-47 with experimental details in SI 5.3. A conclusion of the reactions kinetics experiments is added in the main text as follow:

“The comparable Gibbs free energy barriers of the hydride transfer step and dihydrogen activation step are also consistent with kinetic experiments (SI 5, Table S10

and Figs. 32-47) where 0.5 order in dihydrogen pressure and 1.5 order in iridium concentration were observed.”.

Our gas-uptake facilities only allow maximum reaction pressure of 33 bar and therefore we measured the initial reaction rates referred to Klankermayer’s method for measurement of initial turnover frequencies (DOI: 10.1021/acscatal.9b04977). In typical hydrogenation reactions, substrate coordination is assumed quasi-equilibrium and the hydride-migration-insertion or reductive elimination step is the rate determination step. In either case, a first order in substrate concentration is generally reported for hydrogenation reactions (see refs: Chem. Commun., 2009, 7447-7464; ACS Catal., 9(8), 7535-7547). From DFT data, we did not find any new mechanism regarding reaction kinetics in substrate. We therefore did not perform reaction kinetics experiments in substrate concentration, which we hope does not affect the conclusion.

The authors do show a hydrogen consumption curve in the supp info. However there is a blue as well as a red curve and the authors do not explain what the difference is between the two.

Response: Thanks for your remarks. To make the figure clear, legends are now added. The black line refers to the original reaction pressure observed at specific reaction intervals. The red line simulates the reaction pressure change over time. See an exam below:

$$\frac{n_{sub}}{\Delta_P} = \frac{61.44 \text{ mmol}}{48.0 \text{ bar}} \approx 1.28 \text{ mmol/bar (equation 1)}$$

$$m_n = -m_{\Delta_P} \frac{n_{sub}}{\Delta_P} = 25.2 \frac{\text{bar}}{\text{h}} \cdot 1.28 \frac{\text{mmol}}{\text{bar}} \approx 32.26 \frac{\text{mmol}}{\text{h}} \text{ (equation 2)}$$

$$TOF_{ini} = \frac{m_n}{n_{cat}} = \frac{32.26 \frac{\text{mmol}}{\text{h}}}{0.00004 \text{ mmol}} \approx 806,400 \text{ h}^{-1} \text{ (equation 3)}$$

Figure S1. Analysis of pressure drop curve for anionic Ir-catalyst catalyzed asymmetric hydrogenation of acetophenone (S1) at 80 bar of H₂”

The effect of the amount of solvent is rather bizarre as the product is also a secondary alcohol. Are the authors 100% sure that this effect is not caused by a minor impurity in the isopropanol? Isopropanol can even contain peroxides that could well have caused this effect. It maybe worthwhile to repeat these experiments with rigorously purified isopropanol.

Response: Thanks for your valuable concerns. The amounts of solvent effects on the catalysis were reperformed using rigorously distilled isopropanol, shown in supplementary information, table S5. A higher conversion was observed upon increasing the amount of solvent from 0 to 2 mL. Further increasing the amount of solvent to 4 mL did not give negative or positive effects, implying that some amount of

solvent is required for dissolve both the substrate and the catalyst to obtain excellent reaction rates.

The stability of this catalyst is indeed remarkable. Is it possible to reuse the spent catalyst after a first reaction?

Response: It should be possible as you can see that our catalyst can work well even after 30 days during the 13 million turnover number experiments. After catalytic experiments, the product and solvent should be easily removed under inert atmosphere and as such the spent catalyst can be reused. Unfortunately, our set-ups cannot meet the requirements. In our real industrial productions, we use only up to 10 ppm catalyst. Calculations suggest low benefits for recycling the spent catalyst as huge energy are required to remake the volatiles. The cost due to catalyst is less than 1% according to our calculations.

What is entirely lacking in the manuscript is a summary of the state-of-the-art of the use of anionic complexes in (asymmetric) hydrogenation. Pez already reported the use of anionic complexes in homogeneous hydrogenation of esters in 1981. Then there is of course the work of Fujita, of Rinaldo Poli, and I am sure there will be others as well.

Response: Great suggestion! References are added (40-43). A summary of previous work of using anionic complexes in asymmetric hydrogenation is added in the third paragraph of Introduction as below: *“Inspired by highly reactive anionic reductants and multidentate Noyori-type hydrogenation catalysts, we proposed the integration of the concepts of anionic complexes and multidentate ligands for developing ultra-efficient asymmetric hydrogenation catalysts with high selectivity, stability and reactivity preeminence (Fig. 1C). The characteristic anionic complexes bearing a formal negative charge can, in principle, enable high hydricity and accordingly high catalytic reaction rates, as evidenced from the seminal anionic metal hydride catalysts for hydrogenation reaction of carbonyl compounds by Pez, Poli, and others.”*

The interesting thing about the Pez work is that other later showed that the anionic complexes were readily protonated by the alcoholic solvent. Are the authors 100% sure that this is not also the case here. It would be interesting to measure a MS of the catalyst after or better during hydrogenation to see if the catalyst is still double anionic.

Response: Excellent remarks. HRMS (Figure S18-19) of a solution of Ir-precatalyst and NaOtBu in isopropyl alcohol under 30 bar H₂ showed exact mass of 765.1874 [M-2Na+3H]⁺ and 799.1490 [M-2Na+2H+Cl]⁻, corresponding to protonated and chlorinated Ir-ate catalysts, respectively, in the positive and negative region. Therefore, we believe that we do observe the MS of the anionic Ir-catalyst. Additionally, compared to Pez work, we think that the back protonation is unlikely as the pK_a of our Ir-precatalyst is much lower than the alcohols.

Use of acronyms in the title is usually not allowed.

Response: Done! Title has been changed from “Discovery of an Ir-ate Catalyst for Ultra-efficient Asymmetric Hydrogenation of Ketones with 3S Character (Stable,

Speed and Selectivity)” to “A 13-Million Turnover-Number Anionic Ir-Catalyst for a Selective Industrial Route to Chiral Nicotine”. All “3S” have been deleted and “AH” has been changed to asymmetric hydrogenation to make the reading easier for all readers.

I also do not see the added value of the use of this triple S. I also do not see the added value of the use of this triple S.

Response: Agreed and Corrected!

I would also remove the free commercial for Nicotinell. Most journals do not allow this.

Response: Thanks for your comments. We have removed the picture of Nicotinell from Fig. 3.

Reviewer 2: *In this manuscript, Zhang and coworkers reported a highly efficient asymmetric hydrogenation of ketones catalyzed by an unprecedented Ir-ate catalyst. One of the most impressive aspects is the more than 10 million TON. This is the highest number reported in the field to date, comparable to biocatalytic hydrogen transfer. The reasons for the high activity were analyzed and the ate type Ir complex was found to be the key factor to achieve the excellent performance. This is also an important discovery. On the other hand, this strategy and method have also been used for the efficient synthesis of important chiral compounds including nicotine, and 40 tons of nicotine have been produced, which is also a remarkable achievement. The paper is well prepared, including the relevant theoretical calculation part. Therefore, in view of the innovation and application value of this work, I recommend this work to be published in Nature Communications.*

In addition, some other literatures on highly efficient asymmetric hydrogenation of ketones catalyzed by different transition metals are recommended to be cited, such as: (Ni) Angew. Chem. Int. Ed. 2022, 61, e202115983; (Ir) Nat. Catal. 2020, 3, 621–627; (Mn) Angew. Chem. Int. Ed. 2019, 58, 4973–4977; (Ir) Org. Lett., 2018, 20, 6135-6139; (Pd) Angew. Chem. Int. Ed., 2013, 52, 11632-11636.

Response: Thanks for your kind comments of our work. All the recommended references were cited, as reference 15 16, 25, 33, 34 in the manuscript, which helps to make the manuscript better.

Reviewer 3: *This communication reports on the performance of an iridium catalyst for asymmetric hydrogenation. According to this work, this is already being used in an industrial process for the synthesis of nicotine. The work is closely related to prior work by the authors (cited as refs. 23, 41 and 42, where the last one looks like a pre-print of the present communication, it is not) and it is not entirely clear to me what the focus of this manuscript is meant to be - the catalyst development, the industrial process or the*

accompanying mechanistic postulate and its analysis. I'm thus wondering whether this is of sufficient novelty and significance to the field, and also whether a communication is the correct format - the ESI is extensive and, in my view, necessary to make the paper accessible, such that I would favour presentation as a full paper.

Response: Thanks for your critical comments and concerns, which help us to make clear the focus and novelty. In refs 23 and 41 (now refs 24 and 44), we reported ligand f-amphox based iridium catalyst for asymmetric hydrogenation of aryl-alkyl or alkyl-alkyl ketones up to 1,000,000 TON and >99% ee. In ref 42 (now 45), we reported ligand f-phamidol in asymmetric hydrogenation of acetophenone up to 1,000,000 TON and >99% ee. Ligand f-phamidol was found by serendipity, where f-amphox hydrolyzed during silicon chromatography to give f-phamidol.

In this manuscript, we report an ultra-efficient anionic Ir-catalyst based on f-phamidol for asymmetric hydrogenation of (hetero)aryl-alkyl ketones up to 13 million TON and >99% ee. Based on the orders of improvement of TON up to 1,000,000 (compared to the known highest 10,000 TON) in asymmetric hydrogenation of pyridyl-alkyl ketone and an industrial route for manufacture of chiral Nicotine was presented. The ONa/MH bifunctional mechanism was also firstly presented here. Therefore, we hope the referee can find it reasonable to report this new work and recognize the sufficient novelty and significance to the field. To clarify our point, please see the comparison of the work below:

Previous work

This work

We fully agree with the reviewer that this manuscript should be a research article as commented. For the clarity, we have corrected the title and as well some parts of the main text and supplementary information. Title has been changed from “Discovery of an Ir-ate Catalyst for Ultra-efficient Asymmetric Hydrogenation of Ketones with 3S Character (Stable, Speed and Selectivity)” to “A 13-Million Turnover-Number Anionic Ir-Catalyst for a Selective Industrial Route to Chiral Nicotine”.

The writing is not very clear, with distracting errors of grammar and expression. The abstract is not written in the right style, contains multiple references and some of the information is then repeated in the introduction. In addition, abbreviations are used extensively, affecting the clarity of the writing, and figures appear overloaded and are difficult to follow. These presentational issues distract from the science and will need to be addressed.

Response: Thanks for your valuable and critical comments, which we fully agree with. We have double checked the grammar and expression as far as we can, and have carefully modified the title/abstract/figures and deleted unnecessary abbreviations that are used extensively.

The Title has been changed from “Discovery of an Ir-ate Catalyst for Ultra-efficient Asymmetric Hydrogenation of Ketones with 3S Character (Stable, Speed and Selectivity)” to “A 13-Million Turnover-Number Anionic Ir-Catalyst for a Selective Industrial Route to Chiral Nicotine”.

The abstract is rewritten as “The development of ultra-efficient hydrogenation catalysts for reduction of organic carbonyl compounds is critical for pharmaceuticals, agrochemicals and fine chemicals. However, manufacturing practical catalysts with high selectivity, stability and reactivity preeminence remains unsolved, calling for conceptual advancement. *Herein*, by integration of the concepts of multidentate ligation and anionic complex, we report the first ultra-efficient anionic Ir-catalyst for highly selective construction of chiral alcohols via asymmetric hydrogenation of (nitrogen-containing) ketones. The anionic catalyst features remarkable, biocatalysis-like efficacy of 99% ee (enantiomeric excess), 13,425,000 TON (turnover number) and 224 s⁻¹ TOF (turnover frequency). Quantum chemical studies reveal a novel ONa/MH bifunctional mechanism of the Ir-catalyst. With this anionic Ir-catalyst, a selective industrial route to enantiopure nicotine at 500 kg batch scale has been established, providing 40 tons scale of product.”

Following the reviewer’s comment, previous Fig. 2 has been split into Fig. 2 and Fig. 3, previous Fig. 3 is split into Fig. 4 and Fig. 5, and previous Fig. 4 is split into Fig. 6

and Fig. 7. The Figures in the SI are also reorganized. The abbreviations such as 3S are removed from the manuscript.

The reported work is based on multiple experiments and calculations, but these have not been presented well or logically, making it look like a set of results has been cherry-picked for the communication, while the details have been buried in the ESI. For example, I wondered about the small energy differences calculated between the mono- and diprotonated versions of the catalyst and how these figures were balanced with the sodium base; this has fed into figure 4B, but the details/critical assessment of the results are hard to come by.

Response: Thanks for your valuable and critical comments in terms of the logic and the writing style. We have substantially reorganized the manuscript to improve the presentation and make the main message clear as highlighted in the manuscript and as well in the supplementary information.

We have reorganized the SI into several section. For example, in SI 5, we mainly present the reaction kinetics studies of the anionic Ir-catalyst catalyzed asymmetric hydrogenation of acetophenone, including measurement of initial turnover frequencies, determination of the reaction order in hydrogenation pressure and iridium.

Fig. 4 Formation of anionic Ir-catalyst from Ir-precatalyst. Calculated relative Gibbs free energies at 298.15 K are given in brackets.

Fig. S24. Relative Gibbs free energies (values in kcal mol⁻¹ at 298.15 K) of possible active anionic Ir-catalysts formation under basic condition.

As depicted in Fig. 3A (now Fig 4) and main text discussion in the manuscript, compared to deprotonation of NH function, OH deprotonation is highly energetically favorable, resulting in anionic Ir-catalysts C and D (Fig. S24). Due to the small energy differences between the mono- and deprotonated versions of the catalyst, we consider that Ir-ate catalysts C and D are likely under equilibrium at the experimental condition (excess base). Such Ir-ate complexes were characterized by HRMS, in-situ high-pressure (HP) NMR and ATR-IR spectroscopy as well as DFT calculations. Thus, Ir catalysts A, C and D are all explored completely for acetophenone hydrogenation. Compared to A and anionic Ir-catalyst C (Tables S7), anionic Ir-catalyst D provides energetically favorable pathways involving NNa/MH and ONa/MH bifunctional mechanisms, as shown in Fig. 6 and S26-27. We have deleted the discussion of the role of alkali cations on the hydride transfer step in the manuscript for more clear presentation of the theoretical work.

Similarly, some tests with different numbers of explicit solvent molecules seem to have been performed and are shown in Figure S38, but I struggled to locate a discussion of the different models, how solvent molecules were placed and, again, a critical assessment of the results.

Response: Thanks for your concerns and critical comments. We share the same concerns. "To simplify the calculation of reaction mechanism, we didn't consider the solvated Na⁺ in the model, as with other similar theoretical works (*J. Am. Chem. Soc.* 2014, 136, 3505–3521). However, in real condition, the Na⁺ may be surrounded by the solvent *i*PrOH. Here, considering the possible interactions, including hydrogen bonding

and coordination interactions of solvent molecules with sodium cation, ketone substrate and ligand, two and four *i*PrOH molecules were added involving sufficient interactions between molecules to test the effect on the hydride transfer step. Indeed, the solvated Na^+ has lower polarizability toward the substrate acetophenone, resulting in a relatively higher energy barrier of the hydride transfer step.”

We have added the above detailed description below Fig. S28 (old version in Fig. S38).

Fig. S28 Structures of the transition states of the hydride transfer step upon the active anionic Ir-catalyst D with explicit solvent molecules and the Gibbs free energies

I also wondered whether for this type of study, optimisation in the gas phase followed by calculations of energies in solvation, with a bigger basis set, would be appropriate - at the very least the authors would need to demonstrate that the structural effects are minimal (and I fear they would not be).

Response: We fully agree with this concern. Unfortunately, as numerous optimized structures and corresponding energies need to be calculated in this work, calculations under large basis sets and solvation model for all the cases are not impossible, but are almost impractical because of the time consuming. In fact, we previously also tested the effects of basis set and implicit solvent model on structural optimization and the results showed that the bigger basis set or implicit solvent model gave small effects on optimized structures, implying that the previous results are not fully unreasonable. To show the difference, we have now listed the results below. Table S1 has now been added in Section 2.8 Computational methods of SI.

Table S1. Selected bond length (unit: Å) of species A under different basis sets.

Bond Length	SDD-6-31G*	SDD-6-31+G*	def2-TZVP	SDD-6-31G* (SMD)	def2-TZVP-6-311++G(d,p) (SMD)
Ir-H5	1.692	1.692	1.685	1.693	1.684
Ir-H6	1.682	1.683	1.679	1.690	1.688
Ir-O	2.179	2.180	2.164	2.182	2.175

Ir-N3	2.089	2.087	2.079	2.093	2.082
Ir-N4	2.051	2.054	2.051	2.066	2.063
Ir-P	2.266	2.266	2.244	2.282	2.275

As the reviewer knows, studies that perform structural optimization in the gas phase followed by calculations of energies with a bigger basis set and solvation correction by implicit solvent model have been widely used *ad hoc*. Many times, reasonable results can be provided by such approximate approach, as shown in many research works such as *J. Am. Chem. Soc.* 2021, *143*, 3571–3582; *Nat. Chem.*, 2022, *14*, 1233–1241; *J. Am. Chem. Soc.* 2023, *145*(4), 2305–2314; *J. Am. Chem. Soc.* 2023, *145*(4), 2207–2218. We therefore hope the reviewer can agree that the current approach is not ideal, but practically acceptable.

Calculation results have mainly been presented in terms of energy profiles in the ESI, again with minimal commentary on what was done and why - it would be sensible to expand this into data tables, discussions and the critical evaluations of whether these results can be trusted. For ee's the energy differences are small, such that computational and conformational noise need to be assessed carefully and I find it difficult to fully assess whether the interpretations of these results are reliable.

Response: Thanks for your valuable remarks. Following the suggestions, to make the data better organized and clearer, we have reorganized the related Sections and Figures in SI, and added the detailed description following the corresponding Figures. We have summarized the Gibbs free energy barriers of hydride transfer step and dihydrogen activation step on Ir-catalyst A and anionic Ir-catalyst C and D, as shown in Table S7, which can clearly show that compared to A and anionic Ir-catalyst C, anionic Ir-catalyst D provides energetically favorable pathways involving NNa/MH and ONa/MH bifunctional mechanisms.

Table S7. The Gibbs free energy barriers for Ir-catalyst A and anionic Ir-catalyst C, D.

path	A		C		D	
	ΔG_1	ΔG_2	ΔG_1	ΔG_2	ΔG_1	ΔG_2
OX/MH-R	12.1	33.5	7.8	6.4	6.2	4.7
OX/MH-S	14.9	33.5	7.9	8.4	7.6	5.5
NX/MH-R	10.5	24.5	10.7	17.1	10.8	9.5
NX/MH-S	9.1	24.5	10.6	17.1	10.6	9.3

ΔG_1 and ΔG_2 denote the free energy barrier of the hydride transfer step and the dihydrogen addition step at 298.15K, respectively. X denotes Na or H in corresponding Ir-catalyst A, C and D. Unit: kcal mol⁻¹.

General theoretical methods have been used to calculate the catalytic mechanisms as discussed in last question, and the reaction mechanisms, especially the rate-determining

steps, are decided by the relative Gibbs free energy barriers, where the computational and conformational noise would be cancelled to a large extent, as the systematic errors would be more or less removed when considering the energy difference. Thus, we believe that the computational results of reaction mechanisms are not unreasonable. Considering the complexed solution environment and computational errors, it is difficult to exactly obtain the ee value but qualitative description of the selectivity, which is accord with the experimental results.

Overall, then, there is a lot of work here, but the presentation needs to be improved and the key messages need to be clarified - I'm not convinced this is the right format or indeed the right journal.

Response: Thanks for your concerns and remarks. We have carefully reorganized the manuscript to improve the presentation and make the main messages clear. Detailed modifications can be found in the highlighted manuscript. We hope the reviewer can agree that the research effort and the excellent TON result of this work is worthwhile for an multidisciplinary journal like Nat. Commun.

REVIEWER COMMENTS

Reviewer #1 (Remarks to the Author):

The paper by Zhang and co-workers has much improved. However, I am not satisfied with the kinetics calculations and the discussion of the outcome of the kinetics in relation to the DFT calculations.

The authors discovered a mistake in their calculations and claim that now in both the blue and the red mechanism the reaction with hydrogen is the rate determining. However, taking a closer look at the red curve it now seems that the highest barrier is in the alcohol decomplexation step. This suggests that the reaction is inhibited by the product, which would be quite easy to prove by comparing the rate of a reaction where some product has been added right from the start with a reaction where this is not the case. The barriers of the three basic steps are rather close and so it may be too close to call. But simply stating that the rate determining step is the hydrogen addition is not correct based on this diagram.

The authors have performed the kinetics, unfortunately by measuring the decline of the hydrogen pressure, which is a rather inaccurate way of doing this. I am not suggesting that the authors should redo the experiments under constant pressure. This is a lot of extra work and will not give much more information. However, the authors have made a mistake in the calculation of the order in hydrogen, which is perhaps caused by the fact that the Ln of the hydrogen pressure was the Y-axis and the Ln of the TOF was the x-axis, instead of the other way around. Thus, whereas the authors claim the order in hydrogen is 0.5 it is in reality 2.0, which was already obvious from the raw data. The order in iridium is 1.5. These are not simple kinetics and this does not quite fit the picture that is painted based on the DFT calculations. It will not be easy to get to the bottom of this story unless the calculations contain more mistakes. It will obviously need a lot more work. My proposal would be that the authors correct the obvious mistake with the hydrogen order and admit that the kinetic data do not quite fit the proposed mechanism. They could end with the statement that more research is necessary to be able to propose the definite mechanism.

I personally find the hydrogenation results interesting enough to publish the paper now after these minor changes.

Reviewer #3 (Remarks to the Author):

The calculations on their own are not of the best quality but the authors make a sensible case that, in conjunction with the experimental data and practical considerations, they contribute sufficiently to the insights generated. So, while I maintain that this is not the best format for this work (a full paper in a more specialist journal seems more appropriate), I concede that it has been improved sufficiently to proceed to publication. I would suggest capturing additional parts of the reply to reviewers in the ESI, e.g. the comments around the geometry effects of a better computational approach (despite my misgivings about the length), but the paper is now more balanced.

REVIEWER REPORT
COMMENTS TO AUTHOR:

Reviewer 1:

The paper by Zhang and co-workers has much improved. However, I am not satisfied with the kinetics calculations and the discussion of the outcome of the kinetics in relation to the DFT calculations.

Response: Thanks for your critical remarks and kindly recommendation. We have revised the manuscript and supplementary information to our best. All the corrections were highlighted in yellow color. We must admit that some dispute exists between kinetic experiments and DFT calculations which deserve more works using advanced techniques to gain future insights.

The authors discovered a mistake in their calculations and claim that now in both the blue and the red mechanism the reaction with hydrogen is the rate determining. However, taking a closer look at the red curve it now seems that the highest barrier is in the alcohol decomplexation step. This suggest that the reaction is inhibited by the product, which would be quite easy to prove by comparing the rate of a reaction where some product has been added right from the start with a reaction where this is not the case. The barriers of the three basic steps are rather close and so it may be too close to call. But simply stating that the rate determining step is the hydrogen addition is not correct based on this diagram.

Response: Agreed. Due to the close barriers of the three steps, the stating of the rate determining step is trick. We have corrected 'thus facilitate hydride transfer in the rate and selectivity determining step from' to 'thus facilitate hydride transfer in the selectivity determining step from'. To some extent, we did not expect substrate inhibition as the catalysis reaction was performed excellent well in isopropanol.

The authors have performed the kinetics, unfortunately by measuring the decline of the hydrogen pressure, which is a rather inaccurate way of doing this. I am not suggesting that the authors should redo the experiments under constant pressure. This is a lot of extra work and will not give much more information. However, the authors have made a mistake in the calculation of the order in hydrogen, which is perhaps caused by the fact that the Ln of the hydrogen pressure was the Y-axis and the Ln of the TOF was the x-axis, instead of the other way around. Thus, whereas the authors claim the order in hydrogen is 0.5 it is in reality 2.0, which was already obvious from the raw data. The order in iridium is 1.5. These are not simple kinetics and this does not quite fit the picture that is painted based on the DFT calculations. It will not be easy to get to the bottom of this story unless the calculations contain more mistakes. It will obviously need a lot more work. My proposal would be that the authors correct the obvious mistake with the hydrogen order and admit that the kinetic data do not quite fit the proposed mechanism. They could end with the statement that more research is necessary to be able to propose the definite mechanism.

Response: Agreed. We exchanged X-Y axis of the figure S39. The reaction order in dihydrogen pressure was corrected to 1.9. Given the unusual kinetic orders in both dihydrogen pressure and iridium catalyst, and as well even complexed DFT calculations when considering the solvation, hydrogen bonding, polarization etc. effects, we fully agree with the reviewer that it is rather tricky to give a conclusive mechanism. Corrected as below:

“Unexpectedly, 1.9 order in dihydrogen pressure and 1.5 order in iridium concentration were observed in kinetic experiments (SI 5, Table S10 and Figs. 32-47). Given the unusual kinetic orders in both dihydrogen pressure and iridium catalyst, and as well even complexed DFT calculations when considering the solvation, hydrogen bonding, polarization etc. effects, more research is necessary to be able to propose the definite mechanism.”

I personally find the hydrogenation results interesting enough to publish the paper now after these minor changes.

Response: Thanks. We have made these minor changes.

Reviewer 3: *The calculations on their own are not of the best quality but the authors make a sensible case that, in conjunction with the experimental data and practical considerations, they contribute sufficiently to the insights generated. So, while I maintain that this is not the best format for this work (a full paper in a more specialist journal seems more appropriate), I concede that it has been improved sufficiently to proceed to publication. I would suggest capturing additional parts of the reply to reviewers in the ESI, e.g. the comments around the geometry effects of a better computational approach (despite my misgivings about the length), but the paper is now more balanced.*

Response: Thanks. We have added additional information regarding the computational approach in the ESI 2.7 as “Studies that perform structural optimization in the gas phase followed by calculations of energies with a bigger basis set and solvation correction by implicit solvent model have been widely used *ad hoc*. Many times, reasonable results can be provided by such approximate approach, as shown in many research works such as *J. Am. Chem. Soc.* 2021, 143, 3571–3582; *Nat. Chem.*, 2022, 14, 1233–1241; *J. Am. Chem. Soc.* 2023, 145(4), 2305–2314; *J. Am. Chem. Soc.* 2023, 145(4), 2207–2218. Here, we also tested the effects of basis set and implicit solvent model on structural optimization and the results showed that the bigger basis set or implicit solvent model gave small effects on optimized structures, implying that the calculated results in this work are not fully unreasonable. To show the difference, we have now listed the results below.”.